# Replicability of the EC-Earth3 Earth System Model under a change in computing environment

François Massonnet[1,2], Martin Ménegoz[2,3], Mario Acosta[2], Xavier Yepes-Arbós[2], Eleftheria Exarchou[2], and Francisco J. Doblas-Reyes[2,4]

[1]Georges Lemaître Centre for Earth and Climate Research, Earth and Life Institute, Université catholique de Louvain, 3 Place Louis Pasteur, 1348 Louvain-la-Neuve, Belgium
[2]Barcelona Supercomputing Center (Centro Nacional de Supercomputación), Nexus II-Planta 1 C/ Jordi Girona, 29, 08034 Barcelona, Spain
[3]Institut des Géosciences de l'Environnement, Université Grenoble-Alpes CNRS, CS 40700 38 058 Grenoble Cedex 9, France
[4]Institució Catalana de Recerca i Estudis Avançats (ICREA), Pg. Lluís Companys 23, 08010 Barcelona, Spain

**Correspondence:** François Massonnet (francois.massonnet@uclouvain.be)

**Abstract.** Most Earth System Models (ESMs) are running under different high-performance computing (HPC) environments. This has several advantages, from allowing different groups to work with the same tool in parallel to leveraging the burden of ensemble climate simulations but also offering alternative solutions in case of shutdown (expected or not) of any of the environments. However, for obvious scientific reasons, it is critical to ensure that ESMs provide identical results under changes in computing environment. While strict bit-for-bit reproducibility is not always guaranteed with ESMs, it is desirable that results obtained under one computing environment are at least statistically indistinguishable from those obtained under another environment, which we term a "replicability" condition following the metrology nomenclature. Here, we develop a protocol to assess the replicability of the EC-Earth ESM. Using two versions of EC-Earth, we present one case of non-replicability and one case of replicability. The non-replicable case occurs with the older version of the model and likely finds its origin in the treatment of river runoffs along Antarctic coasts. By contrast, the more recent version of the model provides replicable results. The methodology presented here has been adopted as a standard test by the EC-Earth consortium (27 institutions in Europe) to evaluate the replicability of any new model version across platforms, including for CMIP6 experiments. To a larger extent, it can be used to assess whether other ESMs can safely be ported from one HPC environment to another for studying climate-related questions. Our results and experience with this work suggest that the default assumption should be that ESMs are not replicable under changes in the HPC environment, until proven otherwise.

## 1 Introduction

Numerical models of the climate system are essential tools in climate research. These models are the primary source of information for understanding the functioning of the Earth's climate, for attributing observed changes to specific drivers, and

for the development of mitigation and adaptation policies, among others (IPCC, 2013). Over the years, these models have become more and more complex. Today's Earth System Models (ESMs) consist of several components of the climate system (ocean, atmosphere, cryosphere, biosphere) coupled together, and often feature several millions of lines of code. Due to their high computational requirements, ESMs usually run on high performance computing (HPC) facilities, or supercomputers. As such, climate science now fully entails an important computational component. Climate scientists – developers, users, and now computer scientists, are typically facing three questions regarding computational aspects of the ESMs:

1. How can better performance be achieved for a given model configuration?

2. What is the accuracy of the solution returned by the model (accuracy being defined in this case as how close the simulated output is to the exact solution)?

3. Are the results reproducible under changes in hardware or software?

These three aspects (performance, accuracy and reproducibility) usually conflict and cannot all be achieved at the same time (Corden and Kreitzer, 2015). For example, better performance (1) can be achieved by means of compiler optimization by allowing the compiler to reorder floating-point operations, or even eliminate exceptions (overflow, division by zero). However, this may be the source of non-reproducibility (3) and cause less accuracy in the solution (2). Likewise, achieving bit-for-bit reproducibility (3) is possible but it implies to keep a full control on the flow of operations, which slows down dramatically the execution time of the code (1).

From the three questions listed above, we are primarily interested in the third one because it has received relatively little attention so far from the climate community. While the reproducibility of results should be a natural requirement in science (Berg, 2018), it proves particularly challenging in the context of Earth System Modeling. Users of ESMs have generally a good (or even a full) control on which model source code they use, but they do not always have such a level of control on several constraints external to the model code itself such as: the type of compiler, the version of softwares used, compiler options such as operating system libraries, the maximum number of processors available and the architecture of the cluster itself (among others). It is suspected, however, that such factors might also influence the results produced by the model (see Section 2). Therefore, the meaning of reproducibility for climate sciences, how to test it and whether ESMs provide replicable results are all legitimate questions to ask.

Before we progress with further considerations, it is important to make the distinction between three concepts that are often used interchangeably but represent in fact different notions: repeatability, replicability and reproducibility. Following the Association for Computing Machinery (ACM) we qualify a result as *repeatable* if it can be obtained twice within stated precision by the same experimenter and in the same experimental conditions; we qualify a result as *replicable* if it can be obtained twice within stated precision by different experimenters but in the same experimental conditions; finally, we qualify a result as *reproducible* if it can be obtained twice, within stated precision, by different experimenters in different experimental conditions (https://www.acm.org/publications/policies/artifact-review-badging; Plesser, 2018; McArthur, 2019).

The present study is concerned with the question of replicability of an Earth System Model. That is, it seeks to answer the following question: for the same model, forcings and initial conditions (experimental setup), how do results depend on the

hardware/software constraints, that vary from one experimenter to the next? In particular, we wish to reach three goals with this study:

- We aim at establishing a protocol for detecting the possible non-replicability of an ESM under changes in the HPC environment (compiler environment, distribution of processors, compilation options, flags, ... );

- We aim at reporting examples of non-replicability that highlight the need to run ESMs in full awareness of possible existence of bugs in its code;

- We aim at alerting the climate community about the underestimated role of hardware or software errors in the final model solution.

This article first reviews the existing literature on the topic. Then, we introduce EC-Earth, the ESM used in this work, as well as the protocol for checking its replicability across multiple environments. Finally we report instances of replicability and non-replicability in EC-Earth before formulating recommendations for potential users of climate models.

## 2 Issues of replicability of Earth System Models

It has been long established that output from computer codes of weather, atmospheric and by extension climate models would inevitably face replicability issues. The reason is fundamental. On the one hand, the dynamics underlying the evolution of the atmosphere is highly sensitive to initial conditions as first pointed out by Lorenz (1963). That is, two integrations started from arbitrarily close initial conditions will quickly depart from one another, with doubling time of errors of 2-3 days, due to the chaotic nature of the climate system. On the other hand, computer codes are based on finite-precision arithmetics (which is non-associative and non-commutative), and the representation of numbers or operations can change whenever a different compiler environment, optimization level or Message Passing Interface (MPI) configuration is used. Compounding these two effects inevitably leads to issues of replicability when codes of ESMs are executed in different computing environments, unless specific precautions are taken.

While there are standards in place that are followed by vendors, ESM end-users can be tempted to violate these standards (consciously or not) for the sake of, e.g., better performance. It is in these cases that issues of replicability are the most likely to arise. For example, libraries follow a standard to maintain an order (binary tree sum order is typical) but more aggressive optimizations set up by the user (using the configuration file or by hand) could remove a restrictive order and the more aggressive algorithm could differ between libraries.

We now review in more detail the reasons behind the non-replicability of ESMs and the literature published on the topic so far.

### 2.1 Origins of non-replicability

The governing equations solved by computational models are represented using floating-point variables in binary base. The general representation of a variable consists in several bits to represent the value, exponent and sign of a number. This means

that a finite number of bits are used to represent each real number. This limits the capacity to represent data with enough fidelity, but also determines the magnitude of the numerical errors that will be added to the results of an algorithm due to round-off and other sources of numerical errors. Although there are different reasons for the origin of this kind of errors, most of them are related to (1) how the compiler does the translation of floating point calculations to assembler code when it is trying to optimize it and (2) how the calculations are done during parallel computation. Both of them can follow standards which ensure bit to bit reproducibility, losing some computational performance. However, there are also more aggressive optimizations or approaches which are configured by the user at compilation and could produce reproducibility issues. In the case of Earth System Models, as EC-Earth, they are usually compiled using an external layer and the compilation is transparent to the scientist, who sets up a configuration file per platform, including different keys which decide how aggressive will be the optimizations, thus respecting the standard or not. This means that the compilation options, libraries used or the optimization aggressiveness for different libraries could change between platforms, simply because the configuration file chosen by the user is containing different set ups or because the static libraries have been compiled using different options, where the user is only linking them. These issues are usually negligible, but sometimes could be significant for complex applications. The methodology proposed here will check the results simulated, trying to evaluate if the possible differences could affect the results and forcing the user to check the configuration files in case any anomaly is found. To explain this in detail, reproducibility issues related to compiler optimizations and parallel programming are briefly explained below.

Results produced by a simulation using a specific compiling setup (version, target hardware, flag compilations ...) may be non-replicable under a different compiling setup because trivial round-off errors introduced in the compiled code can potentially trigger significant changes in simulation results. A compiler not only translates the code from a high-level programming language to a low-level language but also tries to improve computational performance of the codes with compiler optimization schemes. The optimizations (the code reorganisation performed by the compiler optimizations before being translated to assembler code) done by the compiler may introduce round-off errors (or even bugs) that are easily overlooked due to the uncertainties or unknowns in ESMs.

Another source of non-replicability is the non-deterministic nature of parallel applications. When global collective communications are used, all the resources working in parallel have to send and receive some data. These data, which are collected by a master process, may arrive in random order (due to delays in message passing between processes) if the user sets up a more aggressive approach – consciously or not – through a configuration file. When data is processed following the order of arrival, the results can end up being non-deterministic because of round-off errors produced by the different order to collect the final result. There are several techniques offered as standards to avoid round-off errors during parallel computation but this implies, in some way, to degrade the computational performance of the execution. This happens, for example, when requiring the collective communications to be sequenced in a prescribed order. Other techniques can be used to reduce the impact of maintaining a particular order to do the operations, such as a binary tree process to calculate the collective communications, avoiding a single sequential order but yet depending on the load balance of the parallel execution to achieve peak performance. All these options can be implemented into the code by the developer, others are inserted directly by the compiler or the library

used for the parallel execution. Again, the configuration depends on the compilation setup chosen and can differ from one HPC environment to another.

## 2.2 State of the art

Rosinski and Williamson (1997) were the first ones to raise the concern of replicability in a global atmospheric model, and formulated several criteria to validate the porting of such models from one computing environment to another. Recognizing already that bit-for-bit replicability would be impossible to meet, they proposed that the long-term statistics of the model solution in the new computing environment should match those in a trusted environment. Subsequent studies tested the sensitivity of results to domain decomposition, change in compiler environment, or usage of different libraries. They all came to the same conclusion that changes in behavior induced by hardware or software differences were not negligible compared to other sources of error such as uncertainty in model parameters or initial conditions. These conclusions were found to hold from weather (Thomas et al., 2002) to seasonal (Hong et al., 2013) and even climate change (Knight et al., 2007) time scales.

Arguably, the most comprehensive and complete study on the topic is that from Baker et al. (2015). Recognizing that the atmosphere exhibits coherency across variables and across space, they proposed a protocol to identify possible non-replicability in standard atmospheric fields, accounting for the strong covariance that may exist between these fields. While useful, the Baker et al. (2015) study addresses only short (1-yr) time scales and is only concerned by atmospheric variables. It is important to acknowledge that variations in hardware or software can potentially impact slower components of the climate system, that the time of emergence of the differences may exceed one year, and that long runs might be needed to disentangle internal climate variability from a true signal. As an example, Servonnat et al. (2013) investigated the replicability of the IPSL-CM5A-LR climate model across several HPC environments. They found that for dynamical variables like surface pressure or precipitation, at least ∼70 years would be needed to ensure that one given signal lies within the bounds of the reference signal.

## 3 Methods

### 3.1 Earth System Model

EC-Earth is a state-of-the-art ESM developed by the EC-Earth consortium, counting close to 20 European institutions (Hazeleger et al., 2011). EC-Earth is a community model developed in a collaborative and decentralized way. EC-Earth consists of coupled component models for atmosphere, ocean, land and sea ice, as described hereunder.

In this study, two versions of the EC-Earth ESM are used. The first one, denoted EC-Earth 3.1 hereafter, is the "interim" version that was developed between the fifth and sixth stages of the Coupled Model Intercomparison Project (CMIP5 and CMIP6). The second one, denoted EC-Earth 3.2, is the "near-CMIP6" version that was used during the two years preceding the official release of EC-Earth for CMIP6.

### 3.1.1 Code information and revisions used

The EC-Earth source codes used for this study were managed through the Subversion (SVN) version control system. For EC-Earth 3.1, the revision r1722 (EC-Earth3.1 official release) of the code was used. For EC-Earth 3.2, the revision r3906 of the code was used.

### 3.1.2 Atmosphere component

The atmosphere component of EC-Earth 3.1 is the Integrated Forecasting System (IFS), cycle 36r4, of the European Centre for Medium-Range Weather Forecasts (ECMWF). IFS is a primitive equation model with fully interactive cloud and radiation physics. The T255 (∼80 km) spectral resolution features 91 vertical levels (up to 1 Pa). The time step is 2700 seconds. IFS is adapted to High Performance Computing (HPC) using the distributed memory paradigm with the standard MPI and the shared memory paradigm using the standard OpenMP. However, the consensus of the EC-Earth community is to use a homogeneous execution for the complete application, taking into account that not all components can use OpenMP. As a result only MPI is used for our tests. It uses domain decomposition to distribute the workload among MPI processes in the horizontal plane, increasing the complexity and overhead of the execution to satisfy the requirements for parallel execution. The atmosphere component of EC-Earth3.2 is the same (IFS cycle 36r4). With respect to the model version used for CMIP5 (Hazeleger et al., 2011), the main updates and improvements in EC-Earth 3.1 include an improved radiation scheme (Morcrette et al., 2008), and a new cloud microphysics scheme (Forbes et al., 2011) in the atmosphere.

### 3.1.3 Ocean and sea ice components

The ocean component of EC-Earth 3.1 is the version 3.3.1 of NEMO (Gurvan et al., 2017). NEMO uses the so-called ORCA1 configuration, which consists of a tripolar grid with poles over northern North America, Siberia and Antarctica at a resolution of about $1°$. Higher resolution, by roughly a factor 3, is applied close to the equator in order to better resolve tropical instability waves. 46 vertical levels are used, and the vertical grid thickness ranges between 6 m and 250 m. The effects of the subgrid scale processes (mainly the mesoscale eddies) are represented by an isopycnal mixing/advection parameterization as proposed by Gent and McWilliams (1990) while the vertical mixing is parameterized according to a local turbulent kinetic energy (TKE) closure scheme (Blanke and Delecluse, 1993). A bottom boundary layer scheme, similar to that of Beckmann and Döscher (1997), is used to improve the representation of dense water spreading. The ocean component is coupled to the Louvain-la-Neuve sea Ice Model, version 3 (LIM3; Vancoppenolle et al., 2009) which is a dynamic-thermodynamic model explicitly accounting for subgrid scale variations in ice thickness. However, in EC-Earth 3.1, only one ice thickness category was used due to numerical instabilities when the default configuration was used with five thickness categories.

EC-Earth 3.2 uses the version 3.6 of the NEMO model and an updated version of the LIM3 model, which this time runs with five ice thickness categories. The ocean grid is identical except that it has 75 vertical levels.

NEMO is adapted to HPC using the shared memory paradigm with the standard MPI. Similar to IFS, it uses domain decomposition to distribute the workload among MPI processes.

### 3.1.4 Land

Both EC-Earth versions 3.1 and 3.2 use the H-TESSEL (TESSEL for Tiled ECMWF Scheme for Surface Exchanges over Land) land surface scheme. H-TESSEL is an integral component of the IFS cycle 36r4, which incorporates land surface hydrology (van den Hurk et al., 2000; Balsamo et al., 2009). It includes up to six land-surface tiles (bare ground, low and high vegetation, intercepted water, and shaded and exposed snow) which can co-exist under the same atmospheric grid-box. The vertical discrimination consists of a four-layer soil that can be covered by a single layer of snow. Vegetation growth and decay are varying climatologically, and there is no interactive biology.

### 3.1.5 Coupling

The atmosphere and ocean–sea ice components of EC-Earth are coupled with the Ocean Atmosphere Sea Ice Soil coupler version 3 (OASIS3; Valcke, 2012). OASIS allows exchanging different fields among components (such as IFS or NEMO) during the execution of EC-Earth. The coupling process involves the transformation of the fields from the source grid to the target grid (including interpolation and conservative operations when it is needed) and the explicit communication among components using MPI communications. OASIS is able to work using MPI to exchange fields between the source and target grids. For EC-Earth 3.1 OASIS3 is used as an independent application, while with EC-Earth 3.2 OASIS3-MCT is called using library functions, thus not requiring dedicated processors.

### 3.2 Protocol for testing replicability

Our protocol (see Fig. 1) is designed to test whether a given version of EC-Earth (either 3.1 or 3.2) gives replicable results under two computing environments, named "A" and "B" for the sake of illustration Proper names will be given in the next section. Additionally, the protocol can be also used to test the replicability of a given version of EC-Earth (3.2 for example) under some specific changes in the computing environment, such as different compilation flags. Before designing the protocol for replicability itself, it was checked and confirmed that both EC-Earth 3.1 and 3.2 are each fully deterministic. This was done using appropriate keys that force the parallel code to be executed in the same conditions, at the expense of an increase in computing time. For each model version, two one-year integrations were conducted under the same computing environment (same executable, same machine, same domain decomposition, same MPI ordering). The results were found to be bit-for-bit identical in both cases. In other words, both EC-Earth 3.1 and 3.2 provided repeatable results.

Our protocol for testing the replicability of EC-Earth entails the use of ensemble simulations. Such ensemble simulations are needed to estimate the magnitude of internally-generated climate variability, and disentangle this variability from actual changes caused by hardware or software differences. In an attempt to reach reasonable statistical power (that is, minimizing the risk of Type-II error or false negatives) while keeping a low significance level (that is, minimizing the risk of Type-I error or false positives; see below), and constrained by limited computational resources, we run 5-member, 20-year simulations for both "A" and "B" computing environments. In the following, each of these 5-member, 20-year ensemble simulations is termed an "integration".

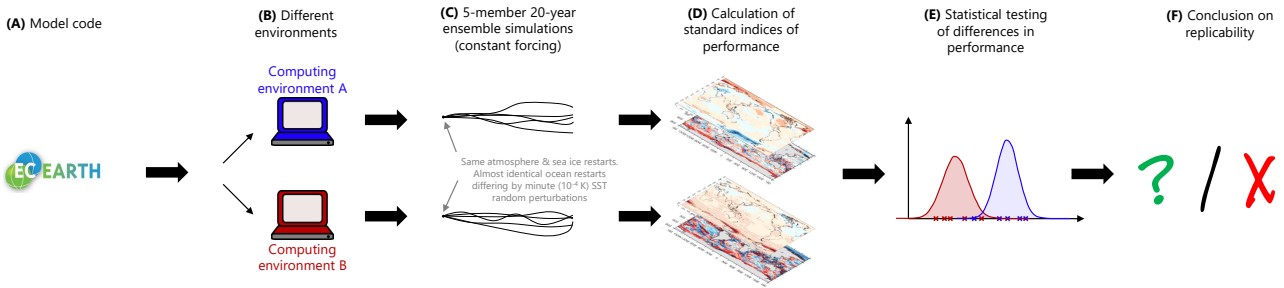

**Figure 1. Protocol for testing the replicability of EC-Earth**. A given model code (A) is ported to two different computing environments (B). Ensemble simulations (C) starting from identical initial conditions are conducted; minute perturbations are added to the ocean restarts to generate ensemble spread. The performance of each ensemble member is computed (D) based on reference reanalysis data and for a number of key variables such as surface air temperature or sea surface salinity. Finally, the null hypothesis that the distribution of performance indices are drawn from the same distribution is tested (E). In case the null hypothesis is rejected at a significance level of 5 % for one variable, the model is deemed non-replicable for that variable (F). Otherwise, the null hypothesis cannot be rejected.

### 3.2.1 Generation of simulations

The five members of the integrations conducted on environments A and B always start from identical atmospheric and sea ice restarts. These restarts are obtained from a long equilibrium simulation conducted at the Italian National Research Council (CNR) (Paolo Davini, personal communication, http://sansone.to.isac.cnr.it/ecearth/init/year1850_tome/15010101/). An ocean

5  restart was also obtained from this equilibrium simulation, and five random but deterministic perturbations were added to the sea surface temperature of this restart at all grid points (gaussian perturbation, standard deviation: $10^{-4}$ K). The introduction of these tiny perturbations allows ensemble spread to develop in integrations A and B and to eventually sample the model's own internal climate variability. Note that by the deterministic nature of the perturbations, pairs of members always start from the same triplet of atmospheric, oceanic and sea ice restarts: the first member of integration A and the first member of integration

10  B start from identical initial conditions, and so for the second member, the third one, etc.

Integrations A and B are conducted under an annually repeating pre-industrial constant forcing. As mentioned above, the integrations are 20-year long. Such a length is considered as a minimum, following Hawkins et al. (2015) who investigated the chance to get a negative trend in annual mean global mean temperature under a 1%/yr increase in CO2 concentration. This study suggests that a negative trend is likely to occur with a probability of 7.8%, 1.2%, and 0.1% when considering respectively

15  10, 15 and 20 years. Similarly, we consider that ensemble experiments starting from slightly perturbed initial conditions could be different after 10 years and are very likely to be similar after 20 years. The initial year is arbitrarily set to 1850, thus the period covered is labelled 1850-1869.

### 3.2.2 Calculation of standard indices

Due to the large amount of output produced by each simulation, it is impossible in practice to compare exhaustively all aspects of integrations A and B to one another. Therefore, the outputs from integrations A and B are first post-processed in an identical way. The code used to post-process the outputs is available (see "Code and data availability") and based on the list of standard metrics proposed by Reichler and Kim (2008). We record for each integration standard ocean, atmosphere and sea ice output variables: 3-D air temperature, humidity and components of the wind; 2-D total precipitation, mean sea-level pressure, air surface temperature, wind stress and surface thermal radiation; 2-D sea surface temperature and salinity, and sea ice concentration. These fields are averaged monthly (240 time steps over 20 years) and the grand time mean is also saved (1 time step over 20 years).

A sensible option would then be to compare together spatial averages of the aforementioned variables from integrations conducted on A and B. However, by definition, spatial averages hide regional differences and one simulation could be deemed replicable with respect to another despite non-replicable differences at the regional scale. To address this point, we rather consider to first compare the fields from each integration at the grid point level to common reference datasets (those used in Reichler and Kim (2008)), and then to compare the resulting metrics from A and B together in order to possibly detect an incompatibility between the two integrations. For each field, a grid-cell area weighted average of the model departure from the corresponding reference is evaluated and then normalized by the variance of that field in the reference data set. Thus, for each field, one metric (positive scalar number) is retained that describes the mismatch between that field in the integration, and the reference field. Five such metrics are available for each integration for each field, since each integration uses five ensemble members.

We stress that the goal of this approach is not to evaluate the quality of the model but rather to come up with a set of scalars characterizing the distance between model output and a reference. Therefore, the intrinsic quality of the reference data sets does not matter for our question. As a matter of fact, the datasets used in Reichler and Kim (2008) and in our protocol correspond to observations affected by historical climate forcings whereas the model output is generated assuming constant pre-industrial forcing. That is, the metrics resulting from the comparison cannot be used in a meaningful way to characterize model quality whatsoever.

### 3.2.3 Statistical testing

For each metric derived in Sec. 3.2.2, two 5-member ensembles need to be compared and it must be determined whether the two ensembles are statistically indistinguishable from one another. Since no prior assumption can be made on the underlying statistical distribution of the samples, we use a two-sample Kolmogorov-Smirnov test (KS test hereinafter). The KS test is non-parametric, which makes it suitable for our application. In the test, the null hypothesis is that the two samples are statistically undistinguishable from each other, that is, drawn from the same distribution.

A Monte-Carlo analysis reveals that for a prescribed level of significance of 5%, the power of the two-sample KS test exceeds 80% (a standard in research) when the means from the two samples are separated by at least two standard deviations,

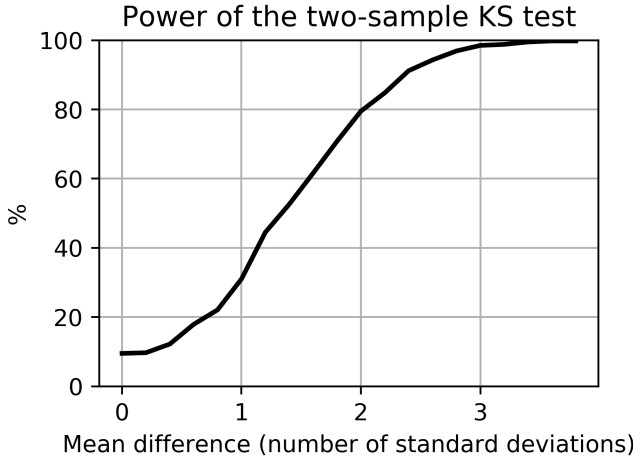

**Figure 2. Power of the statistical test used**. Probability that a two-sample Kolmogorov-Smirnov test (KS test) returns a p-value below a prescribed significance level of 5%, for two 5-member normal samples with equal standard deviations, and means separated by 0, 0.1, 0.2, ...4.0 standard deviations ($x$-axis) (see text for details)

in case of Gaussian distributions (Fig. 2). The probabilities were estimated using 1000 Monte-Carlo runs for each effect size. Gaussian distributions with equal variance were assumed for the samples. Stated otherwise, when the means of two ensembles are separated by less than 2 standard deviations, there is a non-negligible chance ($> 20\%$ at least) that the difference is not detected by the KS test.

### 3.2.4 Experimental setup

For the purpose of this paper, which is to introduce a protocol for replicability and to illustrate cases of (non-)replicability in an ESM, two computing environments were considered (Table 1). Each version of EC-Earth was used to produce one integration in each computing environment, resulting in four experiments (Table 2). The experiments were deployed and run using the Autosubmit scheduler (Manubens-Gil et al., 2016), which ensures an identical treatment of source code, namelist, compilation flags and input files management throughout. It should be noted that each experiment runs under a different domain decomposition, but sensitivity experiments conducted under the same computing environment and where only the domain decomposition was changed, indicated that this did not induce detectable changes in the results (in the sense of the KS test described in Sec. 3.2.3).

Notice in Table 2 that all experiments are using the same compilation flag options for the complete code and external libraries, with the exception of floating point treatment for a0gi and a0g0, in order to prove that the protocol can be used to test different compilation flag options. Experiment a0gi is using "fp-model strict" and a0go is using "fp-model precise". Using fp:strict means that all the rules of standard IEEE 754 are respected and it is used to sustain bitwise compatibility between different

**Table 1.** The two computing environments considered in this study.

| Computing Environment | ECMWF-CCA | MareNostrum3 |
|---|---|---|
| Location | Reading, UK | Barcelona, Spain |
| Motherboard | Cray XC30 system | IBM dx360 M4 |
| Processor | Dual 12-core E5-2697 v2 (Ivy Bridge) series processors (2.7 GHz), 24 cores per node | 2x Intel SandyBridge-EP E5-2670/1600 20M 8-core at 2.6 GHz, 16 cores per node |
| Operating system | Cray Linux Environment (CLE) 5.2 | Linux - SuSe Distribution 11 SP2 |
| Compiler | Intel(R) 64 Compiler XE for applications running on Intel(R) 64, Version 14.0.1.106 Build 20131008 | Intel(R) 64 Compiler XE for applications running on Intel(R) 64, Version 13.0.1.117 Build 20121010 |
| MPI version | Cray mpich2 v6.2.0 | Intel MPI v4.1.3.049 |
| LAPACK version | Cray libsci v12.2.0 | Intel MKL v11.0.1 |
| SZIP, HDF5, NetCDF4 | v2.1, v1.8.11, v4.3.0 | v2.1, v1.8.14, v4.2 |
| GribAPI, GribEX | v1.13.0, v000395 | v1.14.0, v000370 |

compilers and platforms. The "fp:precise" weakens some of the rules in order to introduce some computational optimizations, however it warrants that the precision of the calculations will not be lost.

## 4  Results and Discussion

We first ran two integrations of EC-Earth 3.1 under the ECMWF-CCA and MareNostrum3 computing environments, respectively. Results revealed that for four of the variables considered (out of 13, i.e. about 30%), an incompatibility was detected (Fig. 3). Since the probability of making a Type-I error is set to 5%, the incompatibility might not be explainable by chance only – though we should recognize that all the variables considered display covariances that make the thirteen tests not fully independent. Differences in metrics for sea ice concentration and sea surface temperature appear very large, hinting that more investigation should be devoted to the models behavior at high latitudes.

A spatial analysis of the difference in near surface air temperature (Fig. 5) points the Southern Ocean as the possible region of origin for the discrepancies. From the map, it appears that differences arising from this region could be responsible for the difference seen in all other variables of Fig. 3. We further narrow down the origin of the differences to winter Antarctic sea ice (Fig. 6): September ice extent departs significantly between the two integrations, and the difference in the mean values exceeds by more than a factor of two times the inter-member range of each model. Thus, we can be suspicious about the replicable character of one experiment with respect to another.

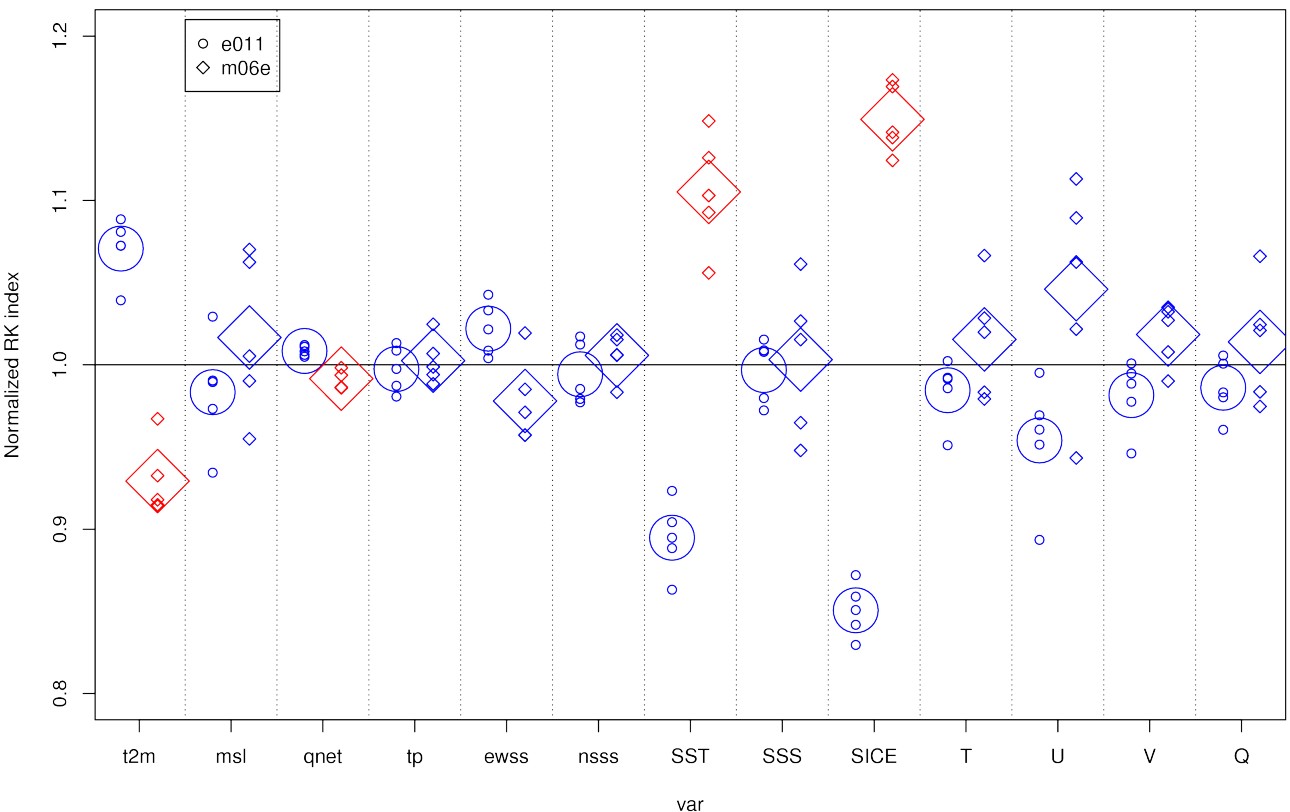

**Figure 3.** Distribution of the normalized Reichler and Kim (2008) metrics for the two simulations e011 (EC-Earth 3.1 on ECMWF-CCA, circles) and m06e (EC-Earth 3.1 on MareNostrum3, squares), for 13 fields: 2-m air temperature (t2m), mean sea level pressure (msl), net thermal radiation (qnet), total precipitation (tp), zonal wind stress (ewss), meridional wind stress (nsss), sea surface temperature (SST), sea surface salinity (SSS), sea ice concentration (SICE), 3-D air temperature (T), 3-D zonal wind (U), 3-D meridional wind (V), specific humidity (Q). The metrics appear in red when the distribution of m06e is statistically incompatible (in the sense of the KS test, see Sec. 3.2.3) with the e011 distribution. The significance level of the KS test is set to 5%.

**Table 2.** The four experiments considered in this study

| Experiment ID | e011 | m06e | a0gi | a0go |
|---|---|---|---|---|
| **Computing Environment** | ECMWF-CCA | MareNostrum3 | ECMWF-CCA | MareNostrum3 |
| **EC-Earth version** | 3.1 | 3.1 | 3.2 | 3.2 |
| **Processors (IFS+NEMO+OASIS)** | 598 | 512 | 432 (288+144) | 416 (288+128) |
| | (480+96+22) | (384+96+22) | (OASIS: library) | (OASIS:library) |
| **F Flags** | -O2 -g | -O2 -g | -O2 -g | -O2 -fp-model |
| | -traceback | -traceback | -traceback -r8 | precise -xHost |
| | -vec-report0 -r8 | -vec-report0 -r8 | -fp-model strict | -g -traceback |
| | -vec-report0 -r8 | -vec-report0 -r8 | -fp-model strict | -g -traceback |
| | | | -xHost | -r8 |
| **C Flags** | -O2 -g | -O2 -g | -O2 -g | -O2 -fp-model |
| | -traceback | -traceback | -traceback | precise -xHost |
| | | | -fp model | -g -traceback |
| | | | strict -xHost | |
| **LD Flags** | -O2 -g | -O2 -g | -O2 -g | -O2 -fp-model |
| | -traceback | -traceback | -traceback | precise -xHost |
| | | | -fp-model strict | -g -traceback |
| | | | -xHost | |
| **Output size** | 141.8 GB | 141.6 GB | 101.3 GB | 101.3 GB |

We then attempted to seek possible physical reasons behind this non-replicability. Investigations led us to detect significant differences in sea surface salinity (SSS) in the Southern Ocean (Fig. 4). The figure allows following the evolution of SSS differences for 1, 5, 10 and 15 days after the initialization from identical ocean and sea ice files. Interestingly, the Southern Ocean is already the scene of differences not seen elsewhere at day 1. The m06e experiment simulates surface waters that are fresher in elongated bands (and suspiciously aligned with the domain's grid) in the eastern Weddell coast and along the Ross ice shelf, but saltier in the rest of the Southern Ocean. This pattern persists through day 15. The lower sea ice areas in m06e than in e011 (Fig.6) are physically consistent with the SSS differences: the additional surface salt in m04e contributes to a weaker vertical oceanic stratification, which could eventually trigger deep convection and lead to systematic differences in winter sea ice coverage. At this stage, it is not clear why m06e has larger SSS already at day 1 away from the coast. Indeed, a salinity anomaly generated at the coast could not propagate equatorward at this speed.

We suspect that these differences in SSS originate from large differences in river runoff values off the coast of Antarctica from one experiment to another. If SSS differences at the coast spread further to the open ocean, they can eventually cause large changes in the ocean column stratification (Kjellsson et al., 2015). If vertical ocean mixing is sufficiently high in one simulation due to large positive sea surface salinity anomalies, it can even prevent sea ice formation in fall and winter. From

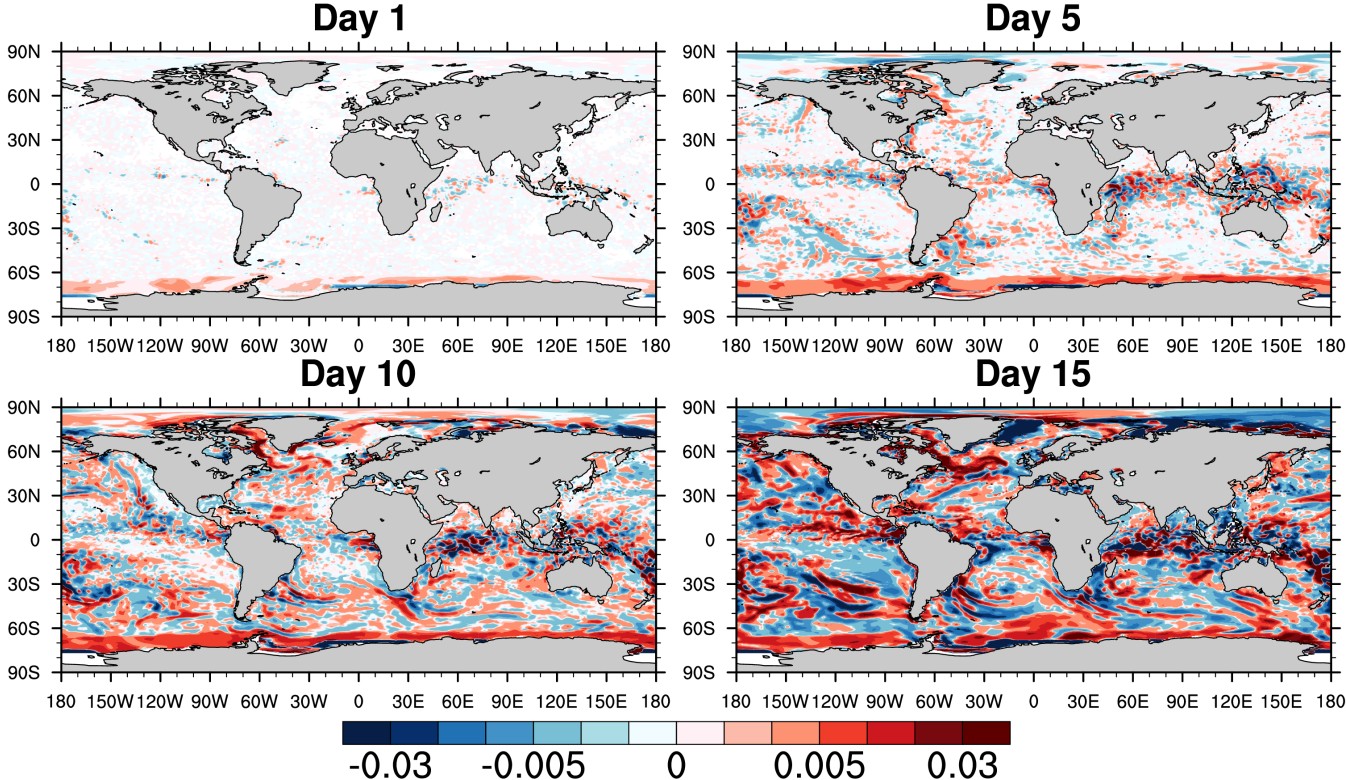

**Figure 4.** Daily mean differences in Sea Surface Salinity (SSS) between ensemble means of experiments m06e and e011 (each has 5 members) on day 1 (January 1st, 1850), 5, 10 and 15 after initialization. Note the non-uniform color scale.

Fig. 4, the problematic simulation seems to be the one carried out on MareNostrum3, although the ECMWF-CCA simulation is also on the low side (the current observed wintertime Antarctic sea ice extent is in the range 15-20 million km²).

The reasons behind differences in river runoff are still to be investigated. We recall that the testing framework is a diagnostic tool that alerts the user to potential issues in model code, but does not identify or fix specific problems. We suspect that a Fortran array involved in the river runoff routines of the NEMO model is not declared in the header of the routine (as it should). When this is the case, the compiler fills the arrays with some default values. However, which default values are set (0.0, 9999.0, NaN...) depends on the compiler itself. We note finally that due to the non-replicability of the results, the output size of the two experiments involved (e011 and m06e) is slightly different (Table 2).

The analysis was then repeated with the newer version of the model, EC-Earth 3.2 (experiments a0gi and a0go on ECMWF-CCA and MareNostrum3, respectively). In that case, we found no instance of incompatibility between any of the 13 variables considered (Fig. 7). A spatial analysis (Fig. 8) suggests that only 1% of the grid points display an incompatibility for 2-m air temperature. We recall that under the null hypothesis of no difference, significant differences are expected to occur 5% of the time. The magnitude of the regional differences in Fig. 8 illustrates the amplitude of climate internal variability, that is larger at middle to high latitudes than in tropical areas. In any case, there is no sufficient evidence to reject the hypothesis that the

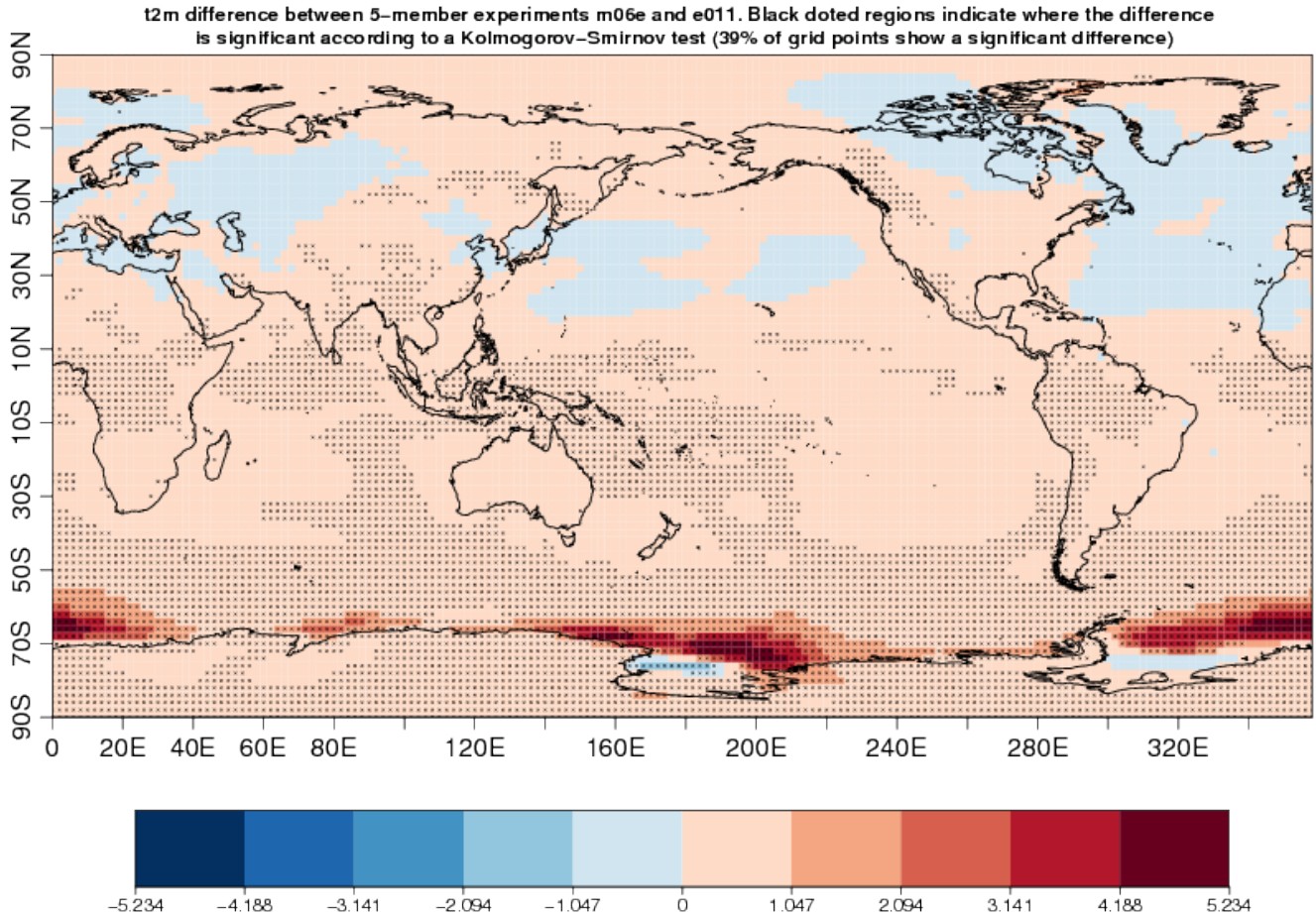

**Figure 5.** Difference in 20-year mean, ensemble mean near surface air temperature between the experiments m06e and e011 (red means m06e is warmer). Dots indicate pixels where the two 5-member samples are statistically incompatible according to the KS test (see Sec. 3.2.3)

two simulations are producing the same climate. Additionally, the results of these two last experiments also suggest that for the compilations flag comparison done here, fp:strict is not needed compared to fp:precise restrictions, producing a similar climate between both experiments (a0gi and a0go) and replicable between identical executions of each experiment. On the contrary, fp:strict increases the execution time of the experiments by 3.6% on ECMWF-CCA and by 3.9% on MareNostrum3, compared to fp:precise option.

The two pairs of figures for the non-replicable case (Figs 3 and 5) and the replicable case (Figs 7 and 8) are useful and compact diagnostics to visually assess the portability of a given model version code to different computing environments. These figures also inherently convey the information that such an assessment is made difficult by the background noise of the climate system. Internal climate variability can obscure real differences caused by hardware changes or, worse, can lead to the

# Sea ice extent S m09

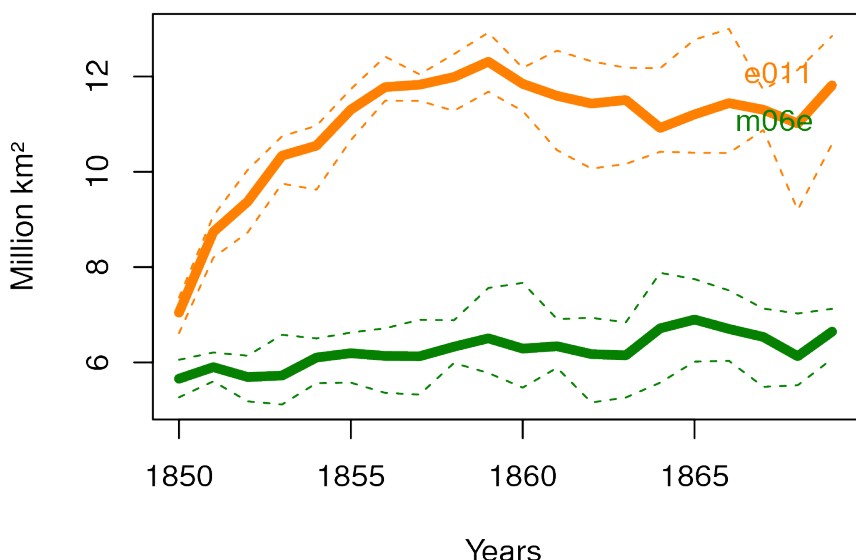

**Figure 6.** Time series of ensemble mean Antarctic September sea ice extent in the two experiments e011 (orange) and m06e (green). The dashed lines indicate the ensemble minimum and maximum (i.e., the range). Sea ice extent is calculated as the sum of areas of ocean pixels containing more than 15% of sea ice.

wrong conclusion that differences exist while the two configurations studied are in fact climatically identical. Nonetheless and despite the small sample size, comparison between the two pairs of figures clearly indicates that one model version (3.1) is not replicable, while the other (3.2) is.

As outlined in an earlier study published in this journal, a successful protocol of replicability should be successful in detect-
5   ing differences arising from known climate-changing modifications (e.g., different physical parameters) but also in detecting differences arising from unknown climate-changing modifications, such as those presented in this paper with the first model version. While the Baker et al. (2015) study did indeed highlight cases of non-replicability with the Community Earth System Model (CESM), these cases were labelled as "borderline" by the authors because of the almost non significance of the differences. In contrast, our first pair of experiments highlights a clear, non-borderline instance of non-replicability with EC-
10   Earth3.1.

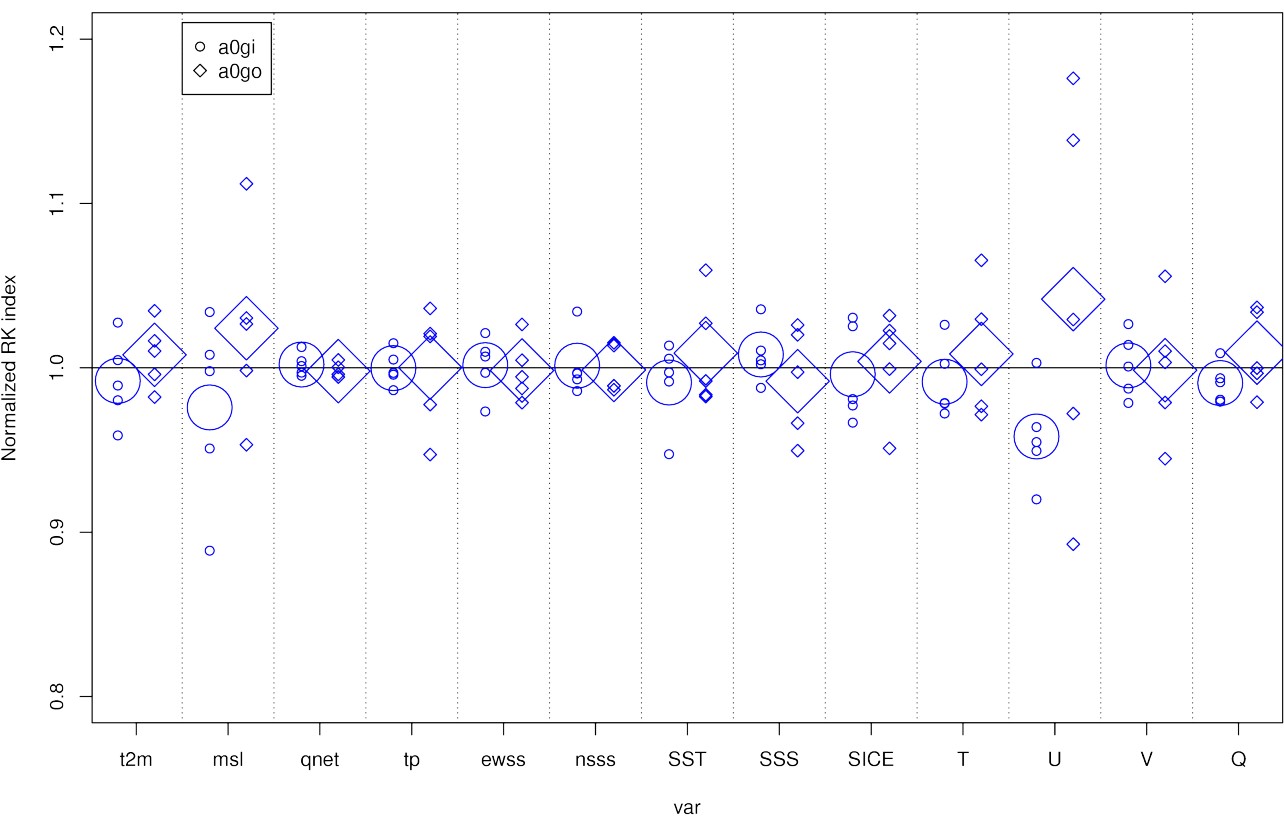

**Figure 7.** Same as Fig. 3 but for the pair of experiments carried out with EC-Earth 3.2, namely a0gi and a0go.

## 5 Concluding remarks and recommendations

Two different versions of the EC-Earth ESM were run under two different computing environments. In one case (model version 3.1), the change of environment implied a significant difference in simulated climates finding its origin in the Southern Ocean, while in the other one (the model version 3.2), it did not, even though one compilation flag option for the treatment of floating point calculations was changed between experiments and proving that the protocol could be used to evaluate different compilation flag options.

What can explain these different outcomes? Our protocol, like others, cannot inform on the source of non-replicability, but can inform whether there may be one (Baker et al., 2015), so in-depth analyses that go beyond the scope of this study would be necessary to trace the origin of non-replicability with version 3.1. However, we suspect that the presence of a bug, present in EC-Earth 3.1 but no longer in EC-Earth 3.2, could explain this result. In fact, we were never able to run the EC-Earth 3.1

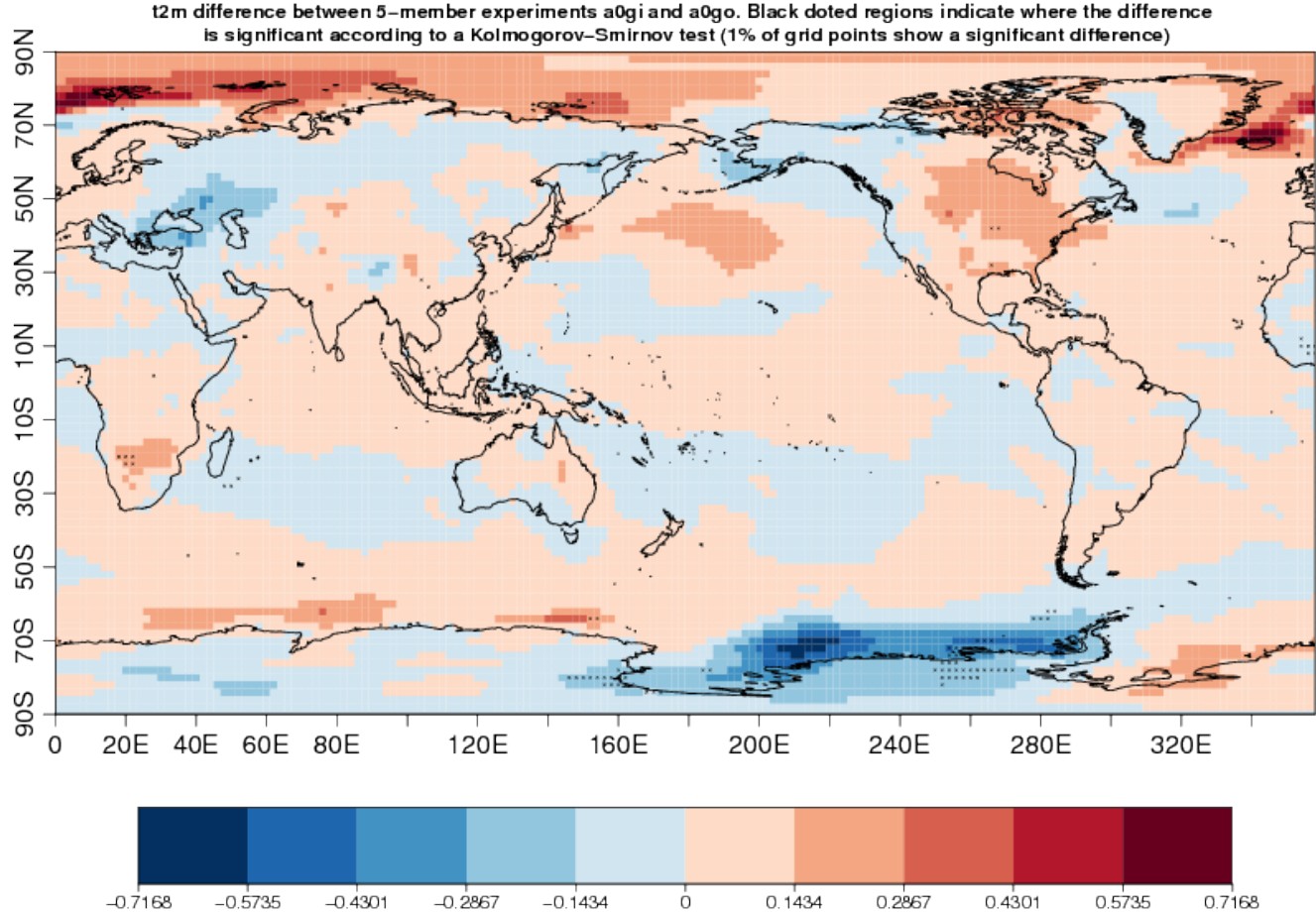

**Figure 8.** Same as Fig. 5 but for the pair of experiments carried out with EC-Earth 3.2, namely a0gi and a0go.

model with the "-fpe0" flag activated during compilation (but could well run the model if this flag was disabled). This flag allows stopping the execution when floating point exceptions, such as division by zero, are encountered. Our guess is that by disabling this flag, the model still encounters bugs (probably linked to the array initialization mentioned in Sec. 4), but these bugs give different outcomes depending on the computing environment. This worrying error that we obtained by porting a climate model from one HPC environment to another one highlights the necessity to choose adequate options of compilation when developing a model, without giving way to the temptation of excluding safe compilation options to bypass a compilation error in a new HPC. Such a result highlights also the fact that porting a code from one HPC to another might be an opportunity to detect errors in model codes.

One of the current limitations in our experimental setup is the fact that EC-Earth code is subject to licensing, and that it is not publicly available for third-party testing. (The protocol for testing its replicability is well publicly available, see "Code and data availability" below). The road to achieve full replicability in climate sciences is, like in other areas of research, full of

obstacles independent of the will of the scientists. The incompatibility between legal constraints and scientific ambitions is one of them (Añel, 2017). Even though the non-accessibility to the software code is a limitation in our study as in others (Añel, 2011) , we still hope that other groups can apply our protocol with their own ESM to confirm our findings, or invalidate them.

Another limitation of the approach is inherent to any statistical testing. While the rejection of the null hypothesis (as with EC-Earth3.1) provides evidence that this hypothesis of replicability is likely not true, the non-rejection of the null hypothesis (as with EC-Earth3.2) does not necessarily imply that it is true – just that it cannot be falsified.

We finally formulate a set of practical recommendations, gained during the realization of this work:

– The default assumption should be that ESMs are not replicable under changes in computing environments. Climate scientists often assume that a model code would give identical climates regardless of where this code is executed. Our experience indicates that the picture is more complicated, and that codes (especially when they are bugged, as they often inevitably are) interfere with computing environments in sometimes unpredictable ways. Thus, it is safer to always assume that a model code will give different results from one computing environment to another, and to try proving the opposite – i.e., that the model executed in the two computing environments gives results that cannot be deemed incompatible. Our protocol fulfills this goal.

– Bugs in models are likely to be interpreted differently depending on the computing environment, and therefore cause significant changes in the simulated climates. In order to herd oneself from this inconvenient situation, and since bugs are by definition hidden to model developers, we formulate the recommendation to (1) systematically compile the model code with the -fpe0 or equivalent flag, so that the model would not be able to run in case of severe bugs, and (2) run systematically the replicability protocol each time the ESM has to be ported to a new machine. From this point of view, protocols for testing climate model replicability such as the one introduced in this article can be viewed as practical tools to spot possible bugs in the code, or to demonstrate at least that a code is not optimally written and developed.

– Besides the frequently quoted sources of prediction uncertainty like climate model error, initial condition errors and climate forcing uncertainty (Hawkins and Sutton, 2009), hardware and software potentially affect the ESM climate (mean state, variability and perhaps response to changes in external forcing, though this latter point was not investigated here) in a way that deserves more attention. Users of climate models have not always the background to appreciate the importance of these impacts. Changes that may appear unimportant, like the reordering of the call to physical routines, could profoundly affect the model results (Donahue and Caldwell, 2018). For climate model users, a better understanding of the conditions that guarantee the replicability of ESMs is a necessary step to bring more trust in these central tools in their work.

– We underline the importance of adopting software development best practices, such as compiling and running climate models without optimizations and with debugging flags prior to launching any production experiment. ESM end-users are primarily interested to simply get the model built and run, either because of time constraints or because of a lack of

technical knowledge. We recommend that these end-users and developers interact more closely to avoid uninformed use of these models by the formers.

## 6 Future work

The cases provided in this study were deliberately highly oriented towards end-users aspects, because the initial motivation for this work was born from a simple practical question: can a coupled ESM simulation be restarted from a different machine without causing climate-changing modifications in the results? Nonetheless, the testing protocol introduced here can be used to address a wider range of questions concerning not only end-users but also developers and computer scientists. A study involving eight institutions and seven different supercomputers in Europe is currently ongoing with EC-Earth. This ongoing study aims to:

- evaluate different computational environments which are used in collaboration to produce CMIP6 experiments; that is can we safely create large ensembles composed of subsets that emanate from different partners of the consortium?

- detect if the same CMIP6 configuration is replicable among platforms of the EC-Earth consortium; that is, can we safely exchange restarts with EC-Earth partners in order to initialize simulations and to avoid long spin-ups?

- evaluate systematically the impact of different compilation flag options; that is, what is the highest acceptable level of optimization that will not break the replicability of EC-Earth for a given environment?

We expect that other groups attempt to test independently the replicability of their own ESM, in order to establish the robustness of our findings obtained with one particular model. In any case, such a practice should become common a standard to gain more confidence in the future use of ESMs.

*Code and data availability.* The code for testing the reproducibility of EC-Earth is available at https://doi.org/10.5281/zenodo.3474777 and also at https://github.com/plesager/ece3-postproc.git. Two sample datasets can be used to test the methodology at http://doi.org/10.23728/b2share.1931aca743f74dcb859de6f37dfad281. Note that the entire code of EC-Earth is not available due to restrictions in the distribution of IFS.

*Author contributions.* All authors contributed to design the study. FM, MM and MA ran the experiments. FM wrote the manuscript and other authors contributed with suggested changes and comments.

*Competing interests.* The authors declare no competing interests.

*Acknowledgements.* We thank Carlos Fernandez Sanchez and three anonymous reviewers for constructive input on this manuscript. We thank Editor Juan Antonio Añel for useful comments and handling our mnuscript. We are grateful to Paolo Davini, Uwe Fladrich, Philippe Le Sager, Klaus Wyser, Ralf Döscher, Etienne Tourigny, Virginie Guemas, Kim Serradell, Oriol Mula -Valls, Miguel Castrillo, Oriol Tinto-Prims, Omar Bellprat and Javier Garcia-Serrano for helpful discussions. We acknowledge the ECMWF and MareNostrum3 for providing computational resources to carry out the experiments. This work was partly supported by the following EU Commission projects: IS-ENES2 (GA 312979), ESiWACE (GA 675191), PRIMAVERA (GA 641727), APPLICATE (GA 727862). It has also received funding from the Ministerio de Economía y Competitividad (MINECO) as part of the HIATUS project CGL2015-70353-R.

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
