# Peer review of "Replicability of the EC-Earth3 Earth System Model under a change in computing environment"

_Geoscientific Model Development, 2019_

## Referee Comment (RC1) · Carlos Fernandez Sanchez (Referee) · 14 Jul 2019

This paper addresses an important topic in Earth System Modelling dealing with the reproducibility of the results when changing the HPC environment (hardware and software). In some way this can derive to how reliable and well written are the models developed. However this is not a new idea in this area. It is well known that different compilers, hardware, options in the compilers, etc... produce different results in the floating point operations, even when using the IEEE754 standard. How significant these results are, is what has to be well analyzed and it is important that the research community is aware of this topic, as it is well presented here.

This paper highlights how a bug or bad coding practice derives in different results depending on the compiler used or flags used in the compiler. Obviously we expect some minor differences in results but not as to being statistically different. Anyway the results show that special care should be taken in the development of the codes and testing of the results. If the hypothesis about the bug in version 3.1 of the code is correct, even two executions of the same model version 3.1 on the same system could provide different results.

The conclusions that "Our results and experience with this work suggest that the default assumption should be that ESMs are not replicable under changes in the HPC environment, until proven otherwise." implies that the codes should be first well tested and evaluated before trying to replicate results in different HPC environments. Anyway testing results in different HPC environments and providing "significant" different results could be a way of demostrating that the code is not well written or developed as presented in this manuscript.

The recommendations about taking care when compiling the code and the flags used (specially the most agressive ones to improve performance) should be extended to other compiler options and special care should be taken before using them. When providing the code and giving results about the code used, a clear description of the version of the code, the system used, compiler version and options used to compile should be provided, along with the operating system used.

The manuscript is very well written and has no major typos or errors

---

## Referee Comment (RC2) · Anonymous Referee #2 · 23 Jul 2019

This paper reviews and develop a protocol to assess the replicability of the EC-Earth Most Earth System Models (ESMs), specifically running two different versions of EC-Earth ESM under two different HPC environments.

Even though the work is very much aimed to EC-Earth3, and takes some assumptions about (generalistic) climate simulations, it reaches its goal of developing an argument about reproducibility. Given this, I just have some minor comments:

[Figure]

**[GMDD]{.orange}**

Interactive
comment

**1   Specific Comments:**

- P2, l22: "influence the outcome of the model". Could you please provide a reference justifying this statement?

- P6, l22: One suggestion: a simple diagram with the different steps of the protocol would support the paper readability and give the reader a quick overview.

- P7, l1: Mentioning "false positive" and "false negative" will improve readability when talking about error types.

- P7, l17: "large amount of output". For more context, could you please specify the size of these outputs?

- P13, l1: Missing "t".

---

## Referee Comment (RC3) · Anonymous Referee #3 · 25 Jul 2019

article

**1   Overall review**

Climate models have a particular problem related to debugging and testing. The underlying dynamics is chaotic, so sensitive to small (floating point roundoff-level) changes. The community has had to extend classical software engineering practice to cases where testing cannot rely on exact bit-for-bit reproducibility. This paper adds to a growing literature on this issue, and this paper cites most of the key papers from that literature.

The particular example chosen here examines two different releases of a widely used climate model EC-Earth. The two versions were tested on two different supercomputers with different hardware, compiler versions, and optimization levels. Comparisons of output were done across a commonly used set of model metrics from Reichler and Kim. The results showed the existence of a possible "uninitialized variable" bug in one version of the model. In the newer version of the model, there is no conclusive evidence of hardware and software causing a changed climate.

The paper is a minor addition to an existing literature, but is useful for forcefully making the case that hardware and software induced answer changes should be very carefully examined as source of differences between two model runs, and the community should systematically adopt rigorous testing processes.

**2  Specific Comments**

- The discussion should mention how the results compare to those in prior perturbation studies mentioned in section 2.2 [e.g., Baker et al. (2015)]

- In section 3.1.4, provide a few more details about the H-TESSEL model (e.g., resolution, grid type, number of vertical levels)

- Section 3.2 should include an explanation of why a 20-year period is necessary to detect code errors that may arise later [e.g., more than 1 year as in Baker et al. (2015)] in the coupled climate model simulations, and whether 20 years is also sufficient for testing different configurations (e.g., active biogeochemistry vs none) or grid resolutions. In particular, how is the 20 years reconciled with the Servonnat et al finding cited at the bottom of page 4, that about 70 years are needed to account for low frequency ocean variability?

- The details about the Monte Carlo simulation in the 2nd sentence of the Figure 1 caption should be moved to paragraph 2 in section 3.2.3.

- The authors do not explicitly compare Figs. 2 and 5 in the text. The authors should likewise explicitly compare Figs. 3 and 6 in the text.

- Figure 3 and Figure 6: The text after the first sentence in the titles should be placed in the caption. The color bars need units, and should have the same numerical range to clarify side-by-side comparison.

- page 12, lines 3-6: Lay readers (e.g., model end-users) may wonder why this bug wasn't fixed. Add a sentence similar to the 3rd sentence in section 5 that emphasizes that the testing framework is a diagnostic tool that alerts the user to potential issues in model code, but does not identify or fix specific problems.

- The null hypothesis that is stated on page 13 line 1 should be introduced in section 3 (methods)

- Page 16: The authors should highlight the importance of adopting software development best practices generally, such as compiling and running climate models without optimizations and debugging flags (e.g., -fpe0) in the second bullet point. In the reviewer's experience, the typical goal of climate model end users is to simply get the model to build and run due to time constraints and/or lack of knowledge about model software and build systems. Users will run simulations with full optimizations only, and may not be aware of issues with code until they examine the output.

- In the conclusions, reiterate that this paper only demonstrates that EC-Earth 3.1 is non-reproducible, not that EC-Earth 3.2 is.

- The data and code availability section should state clearly that the model codes themselves are not publicly available, and therefore a reviewer or reader cannot

independently verify these results. At best they could independently test in a different model to which they may have access.

**3 Grammar / Style**

- Change "hereinafter" to the more common "herein" or "hereafter"

- page 2, line 8: "accuracy" is probably the wrong word, consider changing to "precision or stability"

- Move parenthetical explanation of floating-point math from page 4, lines 8-9 to page 3, line 15, where floating-point math is first mentioned.

- page 7, line 5: "unique and identical" is confusing here, the sentence needs to be rephrased.

- page 10, line 5: "nail down" is used incorrectly; consider changing to "narrow down."

- page 13, line 9: Change "exist" to "be"

---

## Referee Comment (RC4) · Anonymous Referee #4 · 1 Aug 2019

General comments

This paper focuses on the EC-Earth model and considers how to demonstrate it is replicable across different HPC platforms. It adopts a statistical approach and demonstrates the use of metrics with a 'convenient' example of a potential bug in their user code.

The paper does not go into great detail on the causes of differences in the model results on different platforms. In practise, this investigation can be a long and tedious process to isolate the cause either in the user code or sometimes in vendor supplied code.

The technique may be applicable to other models, though it is difficult to tell, given the

complexity of the coupled EC-Earth ESM, what changes would be needed to the authors' methods to achieve reliable results. This paper presents an interesting example on an important issue and publication is recommended following minor revisions.

Specific comments

Page 3, Line 15 (and section 2.1) Sentence beginning 'On the other hand, . . . representation of numbers / operations can ..'. I find this sentence vague. The IEEE standard followed by vendors specifies how numbers are represented and rounded during arithmetic operations, unless a compiler is allowed to go beyond the standard, operations should not be expected to differ. Likewise, why would a different implementation of the MPI library (MPICH .v. OpenMPI) be expected to produce different results if MPI is simply used to move data between tasks? This paragraph appears to loosely suggest differences can arise simply from making these changes. There is no mention of the standards in place that compilers are expected to follow. Compiler options that violate standards are a user choice and often used, as the authors state, to achieve better performance.

Page 3, Lines 16 & 29. Issues in replicability/reproducibility can also arise from the operating system libraries in use, separate from the compiler. For example, optimized vendor supplied versions of the BLAS/LAPACK libraries, often used in ESMs, can give rise to differences compared to other implementations of these libraries. I suggest the authors reword to say 'compiler environment' rather than simply 'compiler' or 'compiler setup' wherever used.

Page 4, lines 2-3. Again, this is rather vague. The user/developer has a great deal of control over what the compiler is allowed to do in terms of optimizing arithmetic operations. To say '(or simply, the translation to assembly code...' is not correct. It is the code reorganisation performed by the compiler optimizations, then translated to assembler, which can be incorrect, either because of user code errors and/or inappropriate compiler options.

Section 3.1.2. The IFS model also supports OpenMP parallelization, can the authors clarify if OpenMP was used?

Section 3.1.4. Is the version of H-TESSEL used part of the IFS CY36, or a different version? Has it been modified from the version supplied with IFS?

Section 3.2. Why 20 years? Does this not depend on the choice of parameters studied?

Section 3.2.2. The IFS model normally outputs GRIB format files, which are a lossy compressed format. Can the authors clarify if they are using output at the precision of the model's arithmetic or some reduced precision format? This is important if looking for small differences and their results?

Page 10, table 2. Several comments: (i) I note that version 3.2 was compiled with the -fp-model strict option which was not used on version 3.1. I would need to check myself but it seems likely to me that this would potentially limit the optimizations the compiler is allowed to perform at -O2. I am curious if the authors think this might be significant for their apparent bug in the river runoff code? (ii) The Mare-Nostrum experiment of 3.2 uses -fp-model precise rather than -fp-model strict. Is this a typo or was it different to the CCA experiment? If so, why? (iii) Can the authors confirm these compiler options were applied to all the code? It is not uncommon to see different compiler flags on selected routines, or compiler directives in the code itself.

Page 12, line 5. It is disappointing that the authors have submitted this paper without completing their investigation into the cause of the disprecancy. If, in the time taken for the paper reviews, the authors are certain the fortran array referred to is the problem, this text should be amended. However, if there has not been any further investigation I would prefer not to see (educated) guesswork on the cause of the problem in the published paper and suggest removing the sentence, as it may turn out to be incorrect. The authors note that version 3.2 did not show the same behaviour. Does this mean that the code they suspect was different between the two model versions? Can the authors clarify in the text whether the offending code was different between the versions?

Corrections

Page 2, Line 29: 'reproducible' should be in italics to match 'replicable' and 'repeatable'.

---

## Editor Comment (EC1) · Juan Antonio Añel (Editor) · 14 Aug 2019

Dear authors,

when submitting a revised version of your manuscript, please, have into account the following issues:

- The information on how to get the exact versions of EC-Earth3 needs to be in the code availability section, including the licence restrictions (it uses IFS so you are not going to be able to archive all the code).

- The reference to the exact version of EC-Earth3 you have used needs to be something

more persistent than an SVN revision marker. If the project ever moved to a different revision control system then it would be impossible to recover the version used.

- The scripts used are presented via a GitHub URL. This is not persistent enough and does not identify the exact version used. The usual way out of this is for authors to use GitHub Zenodo integration to produce an archive version with a DOI. (https://guides.github.com/activities/citable-code/)

- The second sample dataset is referred to by URL instead of DOI, even though there is a DOI for the dataset. This should be fixed.

- DOIs should not be inserted directly in the text. Instead, in line with the Force11 recommendations on data citation, the DOI should occur in a bibliography entry which is cited from the text. (see https://www.geoscientific-model-development.net/about/code_and_data_policy.html).

Juan A. Añel & David Ham

Editorial Board

---

## Author Comment (AC1) · 7 Oct 2019

**Author comments**

Replicability of the EC-Earth3 Earth System Model under a change in computing environment

F. Massonnet et al.

submitted to Geoscientific Model Development, manuscript ID: gmd-2019-91

October 4, 2019

**Editor's comments**

**E1.** Dear authors, when submitting a revised version of your manuscript, please, have into account the following issues: (1) The information on how to get the exact versions of EC-Earth3 needs to be in the code availability section, including the licence restrictions (it uses IFS so you are not going to be able to archive all the code).

**Action.** In the revised version of the manuscript, we have added in the "Code and data availability" section information on how to retrieve the exact versions of EC-Earth used in the article. We also emphasize that not all the code is available due to restrictions in IFS.

**E2.** (2) The reference to the exact version of EC-Earth3 you have used needs to be something more persistent than an SVN revision marker. If the project ever moved to a different revision control system then it would be impossible to recover the version used.

**Action.** We are now moving the non-licensed parts of the EC-Earth code versions that were used in the paper in a Zenodo archive that will be assigned a DOI. We will add this information to the "Code and data availability" section.

**E3.** (3) The scripts used are presented via a GitHub URL. This is not persistent enough and does not identify the exact version used. The usual way out of this is for authors to use GitHub Zenodo integration to produce an archive version with a DOI. (https://guides.github.com/activities/citable-code/)

**Action.** Similarly to the code issue described above, we are now archiving the scripts in a release with DOI, which will be given in the "Code and data availability" section.

**E4.** (4) The second sample dataset is referred to by URL instead of DOI, even though there is a DOI for the dataset. This should be fixed.

Action. This is now fixed in the manuscript.

**E5.** DOIs should not be inserted directly in the text. Instead, in line with the Force11 recommendations on data citation, the DOI should occur in a bibliography entry which is cited from the text. (see https://www.geoscientific-modeldevelopment.net/about/code\_and\_data\_policy.html).

Action. We have taken care to now cite the DOI's as in-line references.

**Referee #1 (Carlos Fernandez Sanchez)**

**R1.1.** This paper addresses an important topic in Earth System Modelling dealing with the reproducibility of the results when changing the HPC environment (hardware and software). In some way this can derive to how reliable and well written are the models developed. However this is not a new idea in this area. It is well known that different compilers, hardware, options in the compilers, etc... produce different results in the floating point operations, even when using the IEEE754 standard. How significant these results are, is what has to be well analyzed and it is important that the research community is aware of this topic, as it is well presented here.

**Reply.** We thank the reviewer Carlos Fernandez Sanchez for evaluating carefully or manuscript and providing constructive input.

**R1.2.** This paper highlights how a bug or bad coding practice derives in different results depending on the compiler used or flags used in the compiler. Obviously we expect some minor differences in results but not as to being statistically different. Anyway the results show that special care should be taken in the development of the codes and testing of the results. If the hypothesis about the bug in version 3.1 of the code is correct, even two executions of the same model version 3.1 on the same system could provide different results.

**R1.3.** The conclusions that "Our results and experience with this work suggest that the default assumption should be that ESMs are not replicable under changes in the HPC environment, until proven otherwise." implies that the codes should be first well tested and evaluated before trying to replicate results in different HPC environments. Anyway testing results in different HPC environments and providing "significant" different results could be a way of demostrating that the code is not well written or developed as presented in this manuscript.

**Reply.** We find the interpretation of the reviewer useful. Presenting the results this way provides another motivation for using a replicability protocol such as described in our work. Finding instances of non-replicability would indeed allow knowing about the possible existence of bugs in the code, and diagnosing where the differences come from would even allow tracing the origin of the bug.

**Action.** We have added a sentence in the conclusion stating that testing for reproducibility is also a way to testing the robustness of the code and the possible existence of bugs.

**R1.4.** The recommendations about taking care when compiling the code and the flags used (specially the most agressive ones to improve performance) should be extended to other compiler options and special care should be taken before using them. When providing the code and giving results about the code used, a clear description of the version of the code, the system used, compiler version and options used to compile should be provided, along with the operating system used.

**Reply.** We agree that a more extended study about compiler flags used could be very interesting. Such investigation could be, on the one hand, on the most aggressive options to improve performance as you suggest but also, on the other hand, on the most restrictive options. However, to do this study correctly, it involves the execution of several experiments using different compiler flags options, library versions, compilers, architectures and model configurations. We are currently working on this study which involves all the EC-Earth community (with 8 participating institutions and 7 platforms around Europe) to run the same experiment (one from CMIP6) under different computational conditions. This implies new authors and results that are oriented to the application of the methodology presented in this paper, so we think that this new work is far from the scope of the present paper and it should be presented in an independent paper. That said, we have added a new section on future work to explain this. Additionally, please notice that the paper results include one example which proves that the methodology can be used to evaluate different compiler flag options. In particular, experiments a0gi and a0go are compiled using -fp-strict and -fp-precise respectively and produce similar results (in the sense of our protocol). Using fp:strict means that all the rules of standard IEEE 754 are respected. fp:strict is used to sustain bitwise compatibility between different compilers and platforms. fp:precise weakens some of the rules, however it warranties that the precision of the calculations will not be lost. On the other hand, fp:strict increases the final execution of the simulation (average of 3 executions) up to 4%. This supports the hypothesis that we produce similar climate results using fp:precise instead of fp:strict, at the benefit of model execution time.

**Action.** All this information has been included in the new version of the paper along with the future work section.

**R1.5.** The manuscript is very well written and has no major typos or errors

Reply. Thanks again.

**Anonymous Referee #2**

**R2.1.** This paper reviews and develop a protocol to assess the replicability of the EC-Earth. Most Earth System Models (ESMs), specifically running two different versions of ECEarth ESM under two different HPC environments.

**R2.2.** Even though the work is very much aimed to EC-Earth3, and takes some assumptions about (generalistic) climate simulations, it reaches its goal of developing an argument about reproducibility.

**Action.** We have added a sentence in the conclusion, emphasizing that the results here are obtained with one model only.

**R2.3.** Given this, I just have some minor comments: P2, I22: "influence the outcome of the model". Could you please provide a reference justifying this statement?

**Reply**. The tone of this sentence was perhaps a bit assertive. We base this notion on the existence of papers listed in Section 2 ("Issues of replicability of Earth System Models").

**Action.** We have reworded the sentence into: "It is suspected, however, that such factors might also influence the results produced by the model (see Section 2)".

**R2.4.** P6, l22: One suggestion: a simple diagram with the different steps of the protocol would support the paper readability and give the reader a quick overview.

Reply. This is a good idea to increase the readability of the paper

**Action.** A new figure presenting the schematics of the method will be added in the revised version of the manuscript

**R2.5.** P7, l1: Mentioning "false positive" and "false negative" will improve readability when talking about error types.

**Action**. We will add "... the risk of Type-I error (occurence of false positives) and Type-II errors (occurence of false negatives)"

**R2.6.** P7, l17: "large amount of output". For more context, could you please specify the size of these outputs?

**Reply.** The output size of each one of the experiments are: e011 (141.8 GB); m06e (141.6 GB); a0gi (101.3 GB); a0go (101.3 GB). There is a slight difference in the output size of experiments e011 and m06e. This is due to the fact that they produce different results, so when netCDF compression is used, the files' size of both experiments also changes.

**Action.** We have added a new row in table 2 with the output size of each one of the experiments and a reference to it when saying "large amount of output". In addition, we now briefly mention why there is a difference in the output size of experiments e011 and m06e.

**R2.7.** P13, l1: Missing "t".

Action. Thanks, this is fixed now.

**Anonymous Referee #3**

**R3.1** Climate models have a particular problem related to debugging and testing. The underlying dynamics is chaotic, so sensitive to small (floating point roundoff-level) changes. The community has had to extend classical software engineering practice to cases where testing cannot rely on exact bit-for-bit reproducibility. This paper adds to a growing literature on this issue, and this paper cites most of the key papers from that literature.

**R3.2.** The particular example chosen here examines two different releases of a widely used climate model EC-Earth. The two versions were tested on two different supercomputers with different hardware, compiler versions, and optimization levels. Comparisons of output were done across a commonly used set of model metrics from Reichler and Kim. The results showed the existence of a possible "uninitialized variable" bug in one version of the model. In the newer version of the model, there is no conclusive evidence of hardware and software causing a changed climate.

**R3.3** The paper is a minor addition to an existing literature, but is useful for forcefully making the case that hardware and software induced answer changes should be very carefully examined as source of differences between two model runs, and the community should systematically adopt rigorous testing processes.

**Reply.** This is indeed the intention of our manuscript, and we are glad that this is well recognized by the reviewer.

**R3.4.** The discussion should mention how the results compare to those in prior perturbation studies mentioned in section 2.2 [e.g., Baker et al. (2015)]

**Reply.** Our understanding from the Baker et al. study is that the compiler-related choices have no detectable influence on the model's climate, that changes in physical parameters lead to successful detection of climate differences (as expected) and that changes in machine or aggressive optimization present borderline cases.

**Action.** We now mention in the discussion that previous studies (e.g., Baker et al. 2015) had already noted a lack of perfect replicability results when the CAM code was ported on different machines, but stress that our case presents a more serious, clearly non-borderline lack of replicability.

**R3.5.** In section 3.1.4, provide a few more details about the H-TESSEL model (e.g., resolution, grid type, number of vertical levels)

**Reply.** We now add more information in the text.

**Action.** We have added the following text: "Both EC-Earth versions 3.1 and 3.2 use the H-TESSEL (TESSEL for Tiled ECMWF Scheme for Surface Exchanges over Land) land surface scheme which incorporates land surface hydrology (van den Hurk et al., 2000, Balsamo et al., 2009). It

includes up to six land-surface tiles (bare ground, low and high vegetation, intercepted water, and shaded and exposed snow) which can co-exist under the same atmospheric grid-box. The vertical discrimination consists of a four-layer soil that can be covered by a single layer of snow. Vegetation growth and decay are varying climatologically, and there is no interactive biology".

New reference:

G. Balsamo, P. Viterbo, A. Beljaars, B. van den Hurk, M. Hirschi, A.K. Betts, K. Scipal, 2009, A revised hydrology for the ECMWF model, *J. Hydro.Met.*, https://doi.org/10.1175/2008JHM1068.1.

**R3.6.** Section 3.2 should include an explanation of why a 20-year period is necessary to detect code errors that may arise later [e.g., more than 1 year as in Baker et al. (2015)] in the coupled climate model simulations, and whether 20 years is also sufficient for testing different configurations (e.g., active biogeochemistry vs none) or grid resolutions. In particular, how is the 20 years reconciled with the Servonnat et al finding cited at the bottom of page 4, that about 70 years are needed to account for low frequency ocean variability?

**Response.** The choice of 20 years is obviously partly subjective, and as pointed by the referee, it is necessary to bring more information about this choice in the manuscript. In climate sciences, it is common to consider a minimum of 20 or 30 years to estimate the mean state of the climate system at the global scale. This is due to the chaotic nature of the climate system that can bring the climate system to different states from similar initial conditions (including small perturbations in model experiments), only because of the internal variability that acts at different timescales, from daily to multi-decadal timescales. The atmosphere acts at higher frequencies than the ocean, possibly explaining why Baker et al. (2015) considered only 1-yr experiments to detect differences in atmosphere-only simulations run on different platforms. We decided to choose a lower boundary of 20 years following Hawkins et al. (2016) who investigated the chance to get a negative trend of the global temperature under a 1% of CO2 increase per year. This study suggests that such a negative trend is likely to occur with a probability of 7.8%, 1.2%, and 0.1% when considering 10, 15 and 20 years of data, respectively. Similarly, we consider that ensemble experiments starting from slightly perturbed initial conditions could be different after 10 years and are very likely to be similar after 20 years. In our protocol, the use of ensemble experiments allows an estimation of the climate internal variability that can be compared to any suspected difference. The choice of 20 years is a tradeoff between the boundary suggested by Hawkins et al. (2016) and the (expensive) computational cost of running 5 member experiments over 20 years (which makes an equivalent of 100 years, more than the 70 years of Servonnat et al. 2016). Obviously, a longer period would be safer to avoid any difference that would not be detected. But we consider that the use of a large set of variables, similarly to what has been done by Servonnat et al. (2016) is helpful to detect the main (ocean, sea ice and atmosphere) issues of replicability.

**Action.** We have added in the text at the end of Section 3.2.1: "As mentioned above, the integrations are 20-year long. Such a length is considered as a minimum, following Hawkins et al. (2016) who investigated the chance to get a negative trend in annual mean global mean temperature under a 1%/yr increase in CO2 concentration. This study suggests that a negative trend is likely to occur with a probability of 7.8%, 1.2%, and 0.1% when considering respectively 10, 15 and 20 years. Similarly, we consider that ensemble experiments starting from slightly perturbed initial conditions could be different after 10 years and are very likely to be similar after 20 years."

**New reference:**

Hawkins E, Smith RS, Gregory JM, Stainforth DA. Irreducible uncertainty in near-term climate projections. *Clim Dyn.* 1 juin 2016;46(11):3807-19.

**R3.7.** The details about the Monte Carlo simulation in the 2nd sentence of the Figure 1 caption should be moved to paragraph 2 in section 3.2.3.

**Action. Done.**

**R3.8** The authors do not explicitly compare Figs. 2 and 5 in the text. The authors should likewise explicitly compare Figs. 3 and 6 in the text.

**Response. Agreed.**

**Action.** We now inter-compare the cases of non-replicability and replicability explicitly in the Section "Results and Discussion"

**R3.9.** page 12, lines 3-6: Lay readers (e.g., model end-users) may wonder why this bug wasn't fixed. Add a sentence similar to the 3rd sentence in section 5 that emphasizes that the testing framework is a diagnostic tool that alerts the user to potential issues in model code, but does not identify or fix specific problems.

**Action. Sentence added.**

**R3.10.** The null hypothesis that is stated on page 13 line 1 should be introduced in section 3 (methods)

**Response.** Indeed, it was missing in the "Methods" section.

Action. A sentence has been added to the "Methods" section.

**R3.11.** Page 16: The authors should highlight the importance of adopting software development best practices generally, such as compiling and running climate models without optimizations and

debugging flags (e.g., -fpe0) in the second bullet point. In the reviewer's experience, the typical goal of climate model end users is to simply get the model to build and run due to time constraints and/or lack of knowledge about model software and build systems. Users will run simulations with full optimizations only, and may not be aware of issues with code until they examine the output.

**Response.** We fully endorse the reviewer's point of view, and agree it should be publicized to the readership of GMD. The point that there is a world of "users" quite disconnected from the world of "developers" is certainly harmful to both parties, because the tools of the latters are often misused by the formers.

**Action.** We now recognize the above point in the conclusion and encourage model users to be more proactive in the computational aspects of models in order to avoid uninformed used of these models.

**R3.12.** In the conclusions, reiterate that this paper only demonstrates that EC-Earth 3.1 is non-reproducible, not that EC-Earth 3.2 is.

Action. Done, a bullet point has been added.

**R3.13.** The data and code availability section should state clearly that the model codes themselves are not publicly available, and therefore a reviewer or reader cannot independently verify these results. At best they could independently test in a different model to which they may have access.

**Response.** This is indeed a critical point that was also raised by the editor.

Action. A sentence has been added to the data and code availability section.

**R3.14.** Change "hereinafter" to the more common "herein" or "hereafter"

Action. Done.

**R3.15.** page 2, line 8: "accuracy" is probably the wrong word, consider changing to "precision or stability"

**Reply.** Thank you for the recommendation. We are trying to follow the consensus in the state of the art where precision is used for the number of bits that the variables has (for example, single (32 bits) or double (64 bits)) and accuracy is used to evaluate how well the climate is simulated, closer to the meaning that we are using here. We have included a brief definition to ensure that the readers understand our meaning.

Action. We have added a definition at the beginning of the text.

**R3.16.** Move parenthetical explanation of floating-point math from page 4, lines 8-9 to page 3, line 15, where floating-point math is first mentioned.

**Action. Done.**

**R3.17.** page 7, line 5: "unique and identical" is confusing here, the sentence needs to be rephrased.

Action. Done, "unique" was deleted.

**R3.18.** page 10, line 5: "nail down" is used incorrectly; consider changing to "narrow down."

Action. The wording has been changed.

R3.19. page 13, line 9: Change "exist" to "be"

Action. Done

**Anonymous Reviewer #4**

**R4.1.** This paper focuses on the EC-Earth model and considers how to demonstrate it is replicable across different HPC platforms. It adopts a statistical approach and demonstrates the use of metrics with a 'convenient' example of a potential bug in their user code.

**R4.2.** The paper does not go into great detail on the causes of differences in the model results on different platforms. In practise, this investigation can be a long and tedious process to isolate the cause either in the user code or sometimes in vendor supplied code.

**R4.3.** The technique may be applicable to other models, though it is difficult to tell, given the complexity of the coupled EC-Earth ESM, what changes would be needed to the authors' methods to achieve reliable results. This paper presents an interesting example on an important issue and publication is recommended following minor revisions.

**Reply.** We thank the reviewer for a constructive input on our manuscript.

**R4.4.** Page 3, Line 15 (and section 2.1) Sentence beginning 'On the other hand, ... representation of numbers / operations can ..'. I find this sentence vague. The IEEE standard followed by vendors specifies how numbers are represented and rounded during arithmetic operations, unless a compiler is allowed to go beyond the standard, operations should not be expected to differ. Likewise, why would a different implementation of the MPI library (MPICH .v. OpenMPI) be expected to produce different results if MPI is simply used to move data between tasks? This paragraph appears to loosely suggest differences can arise simply from making these changes. There is no mention of the standards in place that compilers are expected to follow. Compiler options that violate standards are a user choice and often used, as the authors state, to achieve better performance.

**Reply.** We agree that the sentence is vague and more details should be included to avoid confusion to include details about the standards, how the compilation process for these models works, and which replicability issues could happen with some MPI functions. It is completely true that the different libraries for distributed memory paradigms such as MPICH or OpenMPI follow a standard and the possible differences in optimizations and reproducibility issues are coming from user decisions. However, Earth System Models like EC-Earth are compiled using an external layer and the compilation is transparent to the scientist, who sets up a configuration file per platform, including different keys which decide how aggressive will be the optimizations, respecting the standard or not. This means that the compilation options, libraries used or the optimization aggressiveness for different libraries could change between platforms, simply because the configuration file chosen by the user is containing different set-ups or because the static libraries have been compiled using different options, where the user is only linking them. Additionally, notice that MPI is not only used to move data among subdomains in Earth System Models. Typical MPI arithmetics operations like reductions involve to calculate global numbers from all processes.

If a specific order is not used for these operations, round off differences among runs will happen. As correctly pointed out by the reviewer, libraries follow a standard to maintain an order (binary tree sum order is typical) but more aggressive optimizations set up by the user (using the configuration file or by hand) could remove a restrictive order and the more aggressive algorithm could differ between libraries.

**Action.** All this information has been included in the paper. In particular, we have added a bullet point in the conclusions that the non-replicability is often a consequence of users violating vendor standards (consciously or not...) to achieve better performance.

**R4.5.** Page 3, Lines 16 & 29. Issues in replicability/reproducibility can also arise from the operating system libraries in use, separate from the compiler. For example, optimized vendor supplied versions of the BLAS/LAPACK libraries, often used in ESMs, can give rise to differences compared to other implementations of these libraries. I suggest the authors reword to say 'compiler environment' rather than simply 'compiler' or 'compiler setup' wherever used. Page 4, lines 2-3. Again, this is rather vague. The user/developer has a great deal of control over what the compiler is allowed to do in terms of optimizing arithmetic operations. To say '(or simply, the translation to assembly code...' is not correct. It is the code reorganisation performed by the compiler optimizations, then translated to assembler, which can be incorrect, either because of user code errors and/or inappropriate compiler options.

Action. The wording has been changed.

**R4.6.** Page 4, lines 2-3. Again, this is rather vague. The user/developer has a great deal of control over what the compiler is allowed to do in terms of optimizing arithmetic operations. To say '(or simply, the translation to assembly code...' is not correct. It is the code reorganisation performed by the compiler optimizations, then translated to assembler, which can be incorrect, either because of user code errors and/or inappropriate compiler options.

Reply. We agree.

**Action.** We have extended the sentence to include the reviewer suggestions and improve the understandability of this part.

**R4.7.** Section 3.1.2. The IFS model also supports OpenMP parallelization, can the authors clarify if OpenMP was used?

**Reply.** Although IFS is able to use OpenMP, NEMO is not prepared yet. The good practice in the EC-Earth community is to use a homogeneous execution for the complete MPMD application, using only MPI. Our configurations follow this decision. A brief explanation has been added to the relevant section.

**R4.8.** Section 3.1.4. Is the version of H-TESSEL used part of the IFS CY36, or a different version? Has it been modified from the version supplied with IFS?

**Reply.** We clarify in the text.

**Action.** We have added the sentence: "The land-surface model H-TESSEL is an integral component of the IFS cycle cy36r4. We have not modified it in EC-Earth 3.1 and EC-Earth3.2." Note that a more in-depth presentation of H-TESSEL has been added, following Anonymous Referee #3's request (see comment R3.5).

**R4.9. Section 3.2. Why 20 years? Does this not depend on the choice of parameters studied?**

**Response.** The choice of 20 years is obviously partly subjective, and as also pointed by Anonymous Referee #3, it is necessary to bring more information about this choice in the manuscript. In climate sciences, it is common to consider a minimum of 20 or 30 years to estimate the mean state of the climate system at the global scale. This is due to the chaotic nature of the climate system that can bring the climate system to different states from similar initial conditions (including small perturbations in model experiments), only because of the internal variability that acts at different timescales, from daily to multi-decadal timescales. The atmosphere acts at higher frequencies than the ocean, possibly explaining why Baker et al. (2015) considered only 1-yr experiments to detect differences in atmosphere-only simulations run on different platforms. We decided to choose a lower boundary of 20 years following Hawkins et al. (2016) who investigated the chance to get a negative trend of the global temperature under a 1% of CO2 increase per year. This study suggests that such a negative trend is likely to occur with a probability of 7.8%, 1.2%, and 0.1% when considering 10, 15 and 20 years of data, respectively. Similarly, we consider that ensemble experiments starting from slightly perturbed initial conditions could be different after 10 years and are very likely to be similar after 20 years. In our protocol, the use of ensemble experiments allows an estimation of the climate internal variability that can be compared to any suspected difference. The choice of 20 years is a tradeoff between the boundary suggested by Hawkins et al. (2016) and the (expensive) computational cost of running 5 member experiments over 20 years (which makes an equivalent of 100 years, more than the 70 years of Servonnat et al. 2016). Obviously, a longer period would be safer to avoid any difference that would not be detected. But we consider that the use of a large set of variables, similarly to what has been done by Servonnat et al. (2016) is helpful to detect the main (ocean, sea ice and atmosphere) issues of replicability.

**Action.** We have added in the text at the end of Section 3.2.1: "As mentioned above, the integrations are 20-year long. Such a length is considered as a minimum, following Hawkins et al. (2016) who investigated the chance to get a negative trend in annual mean global mean temperature under a 1%/yr increase in CO2 concentration. This study suggests that a negative trend is likely to occur with a probability of 7.8%, 1.2%, and 0.1% when considering respectively 10, 15 and 20 years. Similarly, we consider that ensemble experiments starting from slightly perturbed initial conditions could be different after 10 years and are very likely to be similar after 20 years."

New reference:

Hawkins E, Smith RS, Gregory JM, Stainforth DA. Irreducible uncertainty in near-term climate projections. *Clim Dyn.* 1 juin 2016;46(11):3807-19.

**R4.10.** Section 3.2.2. The IFS model normally outputs GRIB format files, which are a lossy compressed format. Can the authors clarify if they are using output at the precision of the model's arithmetic or some reduced precision format? This is important if looking for small differences and their results?

**Reply.** In IFS, we output using the reduced precision of GRIB1, which uses 8 and 16 bits per value, depending on the field. In addition, in NEMO we output netCDF data using 32-bit precision. Our tests prove that we can find differences in the simulated climate using our methodology. However, we think that a different study would be necessary to determine which is the minimum output precision to capture small differences in the simulated data, but this is out of the scope of this work.

**R4.11.** Page 10, table 2. Several comments: (i) I note that version 3.2 was compiled with the -fpmodel strict option which was not used on version 3.1. I would need to check myself but it seems likely to me that this would potentially limit the optimizations the compiler is allowed to perform at -O2. I am curious if the authors think this might be significant for their apparent bug in the river runoff code? (ii) The Mare-Nostrum experiment of 3.2 uses -fp-model precise rather than -fp-model strict. Is this a typo or was it different to the CCA experiment? If so, why? (iii) Can the authors confirm these compiler options were applied to all the code? It is not uncommon to see different compiler flags on selected routines, or compiler directives in the code itself.

**Reply.** The reviewer is right. Compilation using fp:strict limits some of the O2 optimizations and it is more limiting (from the computational performance point of view) than fp:precise. We suspected that fp:precise or fp:strict could help to avoid the bug problem with EC-Earth 3.1. However, the results (along with other tests) proved that floating point controls did not solve the problem, these tests are not included in the paper to avoid the extension of the document. On the other hand, the experiments for EC-Earth 3.2 are using fp:precise and fp:strict respectively to prove that the new methodology can be used (as an example) to evaluate the same application for two different flags compilation and platforms. Please check answer to R1.4 for more details, where we prove that we can obtain a similar climate across two platforms and flag compilations options, where fp:strict option is 4% slower in the final execution time, proving that (for this configuration) fp:strict us unneeded compared to fp:precise. All these details have been included in the paper along with a brief explanation about fp:precise and fp:strict. We will extend these results in the future (a future work section has been included for this purpose). About the last comment, the reviewer points out something very important and usually forgotten during compilation for complex models. In our

case, we ensured during our compilation process that all external libraries used the same compilation flags, we have added this information during the model description part.

**Action.** We have added the above information to the paper, also in line with the answer to R1.4 above.

Page 12, line 5. It is disappointing that the authors have submitted this paper without completing their investigation into the cause of the disprecancy. If, in the time taken for the paper reviews, the authors are certain the fortran array referred to is the problem, this text should be amended. However, if there has not been any further investigation I would prefer not to see (educated) guesswork on the cause of the problem in the published paper and suggest removing the sentence, as it may turn out to be incorrect. The authors note that version 3.2 did not show the same behaviour. Does this mean that the code they suspect was different between the two model versions? Can the authors clarify in the text whether the offending code was different between the versions?

**Reply.** We concur with the reviewer that the statement should be supported by more diagnostics; otherwise it might sound loose and unjustified. To this end, we plotted (see figure below) the sea surface salinity (SSS) differences between the experiments m06e and e011, i.e. the two experiments that produce statistically distinguishable results with EC-Earth3.1. The figure allows following the evolution of SSS differences for 1, 5, 10 and 15 days after the initialization from identical ocean and sea ice files. Interestingly, the Southern Ocean is already the scene of differences not seen elsewhere at day 1. The m06e experiment simulates surface waters that are fresher in elongated bands (and suspiciously aligned with the domain's grid) in the eastern Weddell coast and along the Ross ice shelf, but saltier in the rest of the Southern Ocean. This pattern persists through day 15. The lower sea ice areas in m06e than in e011 (Fig. 4 of the original manuscript, also reproduced below for ease of interpretation) are physically consistent with the SSS differences: the additional surface salt in m04e contributes to a weaker vertical oceanic stratification, which could eventually trigger deep convection and lead to systematic differences in winter sea ice coverage. At this stage, it is not clear why m06e has larger SSS already at day 1 away from the coast. Indeed, a salinity anomaly generated at the coast could not propagate equatorward at this speed.

Action. We have added the new figure below in the manuscript and discussed it in the text.

Page 2, Line 29: 'reproducible' should be in italics to match 'replicable' and 'repeatable'.

Action. Done.